# Venous Minus Arterial Carbon Dioxide Gradients in the Monitoring of Tissue Perfusion and Oxygenation: A Narrative Review

**DOI:** 10.3390/medicina59071262

**Published:** 2023-07-06

**Authors:** Arnaldo Dubin, Mario O. Pozo

**Affiliations:** 1Facultad de Ciencias Médicas, Universidad Nacional de La Plata, Cátedras de Terapia Intensiva y Farmacología Aplicada, 60 y 120, La Plata B1902AGW, Argentina; 2Servicio de Terapia Intensiva, Sanatorio Otamendi, Azcuénaga 870, Ciudad Autónoma de Buenos Aires C1115AAB, Argentina; 3Servicio de Terapia Intensiva, Hospital Británico, Perdriel 74, Ciudad Autónoma de Buenos Aires 1280AEB, Argentina; pozomario@gmail.com

**Keywords:** venous minus arterial carbon dioxide partial pressure, cardiac output, tissue perfusion, respiratory quotient, tissue oxygenation

## Abstract

According to Fick’s principle, the total uptake of (or release of) a substance by tissues is the product of blood flow and the difference between the arterial and the venous concentration of the substance. Therefore, the mixed or central venous minus arterial CO_2_ content difference depends on cardiac output (CO). Assuming a linear relationship between CO_2_ content and partial pressure, central or mixed venous minus arterial PCO_2_ differences (P_cv-a_CO_2_ and P_mv-a_CO_2_) are directly related to CO. Nevertheless, this relationship is affected by alterations in the CO_2_Hb dissociation curve induced by metabolic acidosis, hemodilution, the Haldane effect, and changes in CO_2_ production (VCO_2_). In addition, P_cv-a_CO_2_ and P_mv-a_CO_2_ are not interchangeable. Despite these confounders, CO is a main determinant of P_cv-a_CO_2_. Since in a study performed in septic shock patients, P_mv-a_CO_2_ was correlated with changes in sublingual microcirculation but not with those in CO, it has been proposed as a monitor for microcirculation. The respiratory quotient (RQ)—RQ = VCO_2_/O_2_ consumption—sharply increases in anaerobic situations induced by exercise or critical reductions in O_2_ transport. This results from anaerobic VCO_2_ secondary to bicarbonate buffering of anaerobically generated protons. The measurement of RQ requires expired gas analysis by a metabolic cart, which is not usually available. Thus, some studies have suggested that the ratio of P_cv-a_CO_2_ to arterial minus central venous O_2_ content (P_cv-a_CO_2_/C_a-cv_O_2_) might be a surrogate for RQ and tissue oxygenation. In this review, we analyze the physiologic determinants of P_cv-a_CO_2_ and P_cv-a_CO_2_/C_a-cv_O_2_ and their potential usefulness and limitations for the monitoring of critically ill patients. We discuss compelling evidence showing that they are misleading surrogates for tissue perfusion and oxygenation, mainly because they are systemic variables that fail to track regional changes. In addition, they are strongly dependent on changes in the CO_2_Hb dissociation curve, regardless of changes in systemic and microvascular perfusion and oxygenation.

## 1. Introduction

The monitoring of the adequacy of tissue perfusion and oxygenation is a major task in the assessment of critically ill patients. Unfortunately, few tools are available for these goals. The clinical evaluation of skin perfusion by means of the capillary refill time is a valuable method [1]. It is a cheap and easy technique, which can be performed in different sites, such as the fingertip (pulp or nail), earlobe, thumb, forehead, and chest wall. In healthy volunteers, there is a good agreement between capillary refill time measured in the pulp fingertip and the ear lobe [2]. The measurement of capillary refill time, however, is poorly reproducible. It has been suggested that the standardization of the technique might improve its variability [3], but a study showed that even after careful standardization and training, the variability of the method remains wide [4]. The capillary refill time changes according to the environment temperature, age, gender, and skin characteristics [5]. Moreover, skin perfusion could not reflect other relevant microvascular territories [6]. Nevertheless, it gives relevant prognostic information and could successfully guide the resuscitation of patients with septic shock [1,7]. Other technologies aimed at the monitoring of tissue perfusion, such as tissue capnography, are no longer available [8]. The videomicroscopy of sublingual microcirculation is an appealing approach for the direct assessment of tissue perfusion. Despite the fact that different devices are available for this purpose, the present limitations for its clinical utilization are the difficulties in video acquisition and analysis [9].

Global tissue oxygenation has been evaluated through the measurement of blood lactate levels. Hyperlactatemia adequately quantifies the magnitude of tissue hypoxia in low-flow states. In addition, the rate of lactate level reduction, the so-called lactate clearance, might point to the adequacy of resuscitation and the relief of the anaerobic metabolism. On the other hand, increased or persistently high levels of lactate might also express the activation of aerobic glycolysis in hypermetabolic states, such as sepsis. [10]. Thus, it could be a misleading goal for resuscitation [7]. In experimental models of oxygen supply dependency, the abrupt rise in the respiratory quotient (RQ) indicates the start of anaerobic metabolism [11,12,13,14]. Regrettably, the metabolic carts needed for the measurement are not commonly used in ICUs.

Given the limitations associated with the measurement of lactate, venous minus arterial carbon dioxide partial pressure difference (P_v-a_CO_2_) and its ratio to arterial minus venous oxygen content (P_v-a_CO_2_/C_a-v_O_2_) have been proposed for the monitoring of tissue perfusion and oxygenation, as surrogates of tissue minus arterial PCO_2_ difference (P_t-a_CO_2_) and RQ, respectively [15]. For these purposes, mixed or central venous samples have been used (P_mv-a_CO_2_, P_mv-a_CO_2_/C_a-mv_O_2_, P_cv-a_CO_2_, and P_cv-a_CO_2_/C_a-cv_O_2_, respectively). This review aimed to comprehensively discuss the physiological determinants, as well as the experimental and clinical evidence, supporting the usefulness and limitations of both variables for the monitoring of critically ill patients.

## 2. Venous Minus Arterial Carbon Dioxide Partial Pressure Difference

### 2.1. Physiological Background

CO_2_ is an important side product of both glycolysis and the Krebs cycle. The CO_2_ production (VCO_2_) is proportional to the magnitude of the oxidative metabolism. During states of tissue hypoxia related to reductions in oxygen transport (DO_2_), the aerobic VCO_2_ decreases as a result of the depressed oxidative metabolism, but the anaerobic VCO_2_ ensues because of the bicarbonate buffering of anaerobically generated protons. Following its concentration gradient, the CO_2_ diffuses from the sites of production in the mitochondria and the cytosol into the extracellular space and the capillaries. In this way, the PCO_2_ of ~40 mmHg on the arterial side increases to ~45 mmHg on the venous side of the capillaries. Thus, there is a positive venous minus arterial carbon dioxide content difference (C_v-a_CO_2_). It results in P_mv-a_CO_2_ and P_cv-a_CO_2_ values that normally range from 2 to 6 mm Hg. It is worthy of note that the CO_2_ is transported in the blood in three different forms: physically dissolved (10%), as bicarbonate (80%), or bound to Hb as carbamate (10%). The proportion of these forms can be substantially changed by different factors [16].

According to Fick’s principle, systemic VCO_2_ is the product of cardiac output (CO) multiplied by C_v-a_CO_2_ [17]. Consequently, C_v-a_CO_2_ is directly proportional to VCO_2_ and inversely proportional to CO. The changes in VCO_2_ modify the ability of CO_2_ gradients to track the alterations in blood flow. In hypothermia, the tissue hypoperfusion induced by hemorrhagic shock does not increase the intestinal mucosal P_t-a_CO_2_ because of the reduction in the VCO_2_ [18].

Another problematic issue related to the clinical usefulness of Fick’s principle applied to CO_2_ for the monitoring of blood flow is the measurement of CO_2_ content. Determination by direct tonometry is extremely cumbersome. On the other hand, the calculation of CO_2_ content depends on complex formulae that frequently produce unacceptable errors. The method more commonly used was allegedly validated in comparison with manometric measurements performed by the Van Slyke method [19]. The authors found an excellent correlation between both determinations. Even though, using data provided in the manuscript, it is possible to calculate the 95% limits of agreement between calculated and measured CO_2_ content. The resulting value is 4.66 mL/100 mL, which is very wide. Thus, the methods are not interchangeable, especially considering the error propagation related to the calculation of C_v-a_CO_2_. Accordingly, 5–10% of the calculated C_v-a_CO_2_ values are negative, which is not physiologically possible. Improved algorithms for the calculation of CO_2_ content have been developed, but they still show inaccuracies [20,21].

Taking into account these drawbacks, P_v-a_CO_2_ is commonly used instead of C_v-a_CO_2_. The relationship between CO_2_ content and partial pressure, however, is not straightforward and depends on several factors (Figure 1):

(1) Position on the CO_2_Hb dissociation curve: Given the curvilinear characteristics of the curve, the relationship between CO_2_ partial pressure and content varies over the entire range of values. In the steeper portion (low PCO_2_), the increases in PCO_2_ at any CO_2_ content are smaller than in the flattened part (high PCO_2_).

(2) Haldane effect: Oxygenated Hb has a lower capacity for CO_2_ binding. In this way, similar CO_2_ content is associated with higher PCO_2_ at higher oxygen saturations [22]. This mechanism favors the Hb loading of CO_2_ produced by the tissue metabolism in the peripheral capillaries and its unloading in the lungs. Although the PCO_2_ only falls from 45 mmHg on the venous side to 40 mmHg on the arterial side, the CO_2_ content decreases by a much greater extent (Figure 1).

(3) Effect of acidosis: Metabolic acidosis decreases the Hb ability to transport CO_2_ [23].

(4) Hemodilution: Anemia produces higher PCO_2_ values because of the reduced Hb binding [24].

(5) Temperature: Increases in temperature induce a right shift in the HbCO_2_ dissociation curve [25].

Considering these mechanisms, P_v-a_CO_2_ and P_t-a_CO_2_ not only depend on blood flow and VCO_2_ but also on changes in the CO_2_Hb dissociation curve (Figure 2). Shifts in the CO_2_Hb dissociation curve can induce major changes in those differences.

Another relevant concept is that CO_2_ gradients are determined by flow, not by DO_2_. Despite similar degrees of oxygen supply dependence in isolated hindlimbs, regional P_v-a_CO_2_ increased more than twofold in ischemic hypoxia and remained unchanged in hypoxic hypoxia, in which blood flow is normal [26]. Similar findings were described in whole animal models of hypoxic and anemic hypoxia, in which not only systemic and regional P_v-a_CO_2_ but also P_t-a_CO_2_ failed to reflect tissue hypoxia [27,28,29]. In both situations, blood flow is preserved. Therefore, CO_2_ differences depend on flow, and not on tissue hypoxia.

### 2.2. Venous Minus Arterial Carbon Dioxide Partial Pressure in Shock States

During the reductions in CO, there are opposite changes in O_2_ and CO_2_ venous content. Low-flow states are characterized by low venous O_2_ saturation and high venous PCO_2_. In low CO states, tissue and venous hypercarbia are ubiquitous phenomena that arise as a consequence of the reduced washout of CO_2_. In the eighties, the occurrence of venous hypercarbia during cardiac arrest was well-documented [30,31,32]. Experimental and clinical studies also found a widened P_v-a_CO_2_ in other low CO states, such as hemorrhagic shock [33,34,35] and cardiac failure [32]. In hemorrhagic shock, P_v-a_CO_2_ predictably reflects changes in CO. In acute progressive bleeding, the reductions in CO induce semilogarithmic increases in P_mv-a_CO_2_ (Figure 3) [28]. This regression fitting was repeatedly found in several conditions [36,37,38].

In experimental endotoxemic models and in patients with septic shock, P_v-a_CO_2_ also tracks changes in CO [37,39,40,41,42,43]. In the different studies, the strength of the correlation between P_v-a_CO_2_ and CO was quite variable. For example, an observational study in septic patients found a weak but significant correlation between P_cv-a_CO_2_ and CO (R^2^ = 0.07, *p* < 0.0001) [42]. Nevertheless, the proper surrogate for CO is P_cv-a_CO_2_, not P_mv-a_CO_2_. The same study showed a poor agreement between P_cv-a_CO_2_ and P_mv-a_CO_2_ (95% limits of agreement = 9 mmHg), which is similar to that reported elsewhere [44]. Therefore, the variable strength of the correlation between P_v-a_CO_2_ and CO could be explained by either modification in the other determinants (VCO_2_ and HbCO_2_ dissociation curve) or the use of P_cv-a_CO_2_ instead of P_mv-a_CO_2_. In spite of this, P_cv-a_CO_2_ and P_mv-a_CO_2_ depend on CO. This expression of Fick’s principle applied to CO_2_ was confirmed in systematic reviews including large numbers of critically ill and septic patients [45,46].

Given that low values of P_cv-a_CO_2_ were associated with an improved outcome, it has been suggested as a goal for resuscitation [41,43,45,46,47,48,49,50]. Yet, its usefulness for this purpose has never been confirmed. On the contrary, a small, controlled study showed that resuscitation aimed to improve P_cv-a_CO_2_ increases mortality [51].

As a relevant conclusion, P_cv-a_CO_2_ and P_mv-a_CO_2_ are strongly dependent on CO in physiological conditions and in shock states, including septic shock. Nevertheless, the ability of these variables to track CO is dampened by many factors:

(1) Haldane effect: When venous oxygen saturation increases as the result of increased blood flow, changes in venous blood CO_2_ partial pressure and content may differ from each other because of the Haldane effect [52]. In patients with septic shock, dobutamine-induced changes in CO were not followed by decreases in P_mv-a_CO_2_ because of the simultaneous increase in venous O_2_ saturation [44].

In hyperoxia, the Haldane effect also determines increases in P_cv-a_CO_2_ [53], even in the absence of changes in systemic and microvascular hemodynamics [54].

(2) Metabolic acidosis: The right shift in the HbCO_2_ dissociation curve [23] produces greater increases in PCO_2_ on the venous than on the arterial side. Therefore, metabolic acidosis can significantly increase P_v-a_CO_2_ regardless of any change in blood flow [29,44,55].

(3) Hemodilution: Anemia also affects the ability to transport CO_2_. As repeatedly shown, hemodilution is associated with opposite changes in C_v-a_CO_2_ and P_v-a_CO_2_: C_v-a_CO_2_ decreases and P_v-a_CO_2_ increases (Figure 4) [28,29].

(4) Acute changes in ventilation: P_mv-a_CO_2_ increases with hyperventilation and decreases with hypoventilation [52,56,57]. Underlying mechanisms might be the reduction in blood flow and the increase in VCO_2_ driven by systemic alkalosis [58].

(5) Changes in temperature: Changes in body temperature induce parallel modifications in oxidative metabolism and VCO_2_ [18].

(6) Use of central instead of mixed venous samples: There are wide 95% limits of agreement between calculations of P_v-a_CO_2_ using central or mixed venous blood [42,44]. Thus, P_cv-a_CO_2_ might not reflect CO as well as P_mv-a_CO_2_.

(7) The variability of the measurements: Given the variability of the measurements in successive determinations of the P_v-a_CO_2_ gap, it is recommended to consider only variations of at least ±2 mmHg as real changes [59].

### 2.3. Venous Minus Arterial Carbon Dioxide Partial Pressure as a Monitor of Microcirculatory Perfusion in Septic Shock

Septic shock is a condition in which the coherence between systemic hemodynamics and microcirculation can be lost. A systemic hyperdynamic state can coexist with microvascular hypoperfusion in some territories. Tissue hypoperfusion could be identified by means of P_t-a_CO_2_. Accordingly, experimental and clinical studies showed that sublingual, intestinal mucosal, and cutaneous P_t-a_CO_2_ correlate with the respective microcirculatory flow [60,61,62]. In contrast, the systemic P_v-a_CO_2_ depends on CO, while the regional P_v-a_CO_2_ of different organs is determined by the corresponding blood flow of each organ. In conditions characterized by the dissociation between systemic cardiovascular variables and microcirculation, systemic P_v-a_CO_2_ is also dissociated from P_t-a_CO_2_ and microcirculation. Thus, systemic variables, such as P_mv-a_CO_2_ and P_cv-a_CO_2_ could fail to reflect tissue hypoperfusion. Nevertheless, many reviews recommended the use of P_cv-a_CO_2_ for the monitoring of microcirculation in critically ill patients, even in situations of normal or high CO [15,49,63,64,65,66,67]. This recommendation is only based on the results of an observational study, which assessed the relationship of P_mv-a_CO_2_ to systemic hemodynamics and sublingual microcirculation [66]. Seventy-five patients with septic shock were evaluated at basal conditions and 6 h later. The study showed that changes in P_mv-a_CO_2_ correlated with changes in the proportion of perfused microvessels, but there was no such correlation between P_mv-a_CO_2_ and CO. The main conclusion of the study was that P_mv-a_CO_2_ could reflect microvascular flow and not systemic hemodynamic variables. Considering that this suggestion challenges Fick’s principle, the lack of correlation between P_mv-a_CO_2_ and CO should have been explained by changes in the many other determinants of P_mv-a_CO_2_, mainly those that modify the dissociation of CO_2_ from Hb. The authors stated that corrections for the Haldane effect were done, but this point was not clearly addressed in the manuscript, especially because O_2_ saturations were calculated instead of being directly measured by a co-oximeter.

Another study, performed in patients with cardiogenic shock on venoarterial extracorporeal membrane oxygenation, found that P_v-a_CO_2_ was higher in nonsurvivors than in survivors (7.4 mm Hg [5.7–10.1] vs. 5.9 mm Hg [3.8–9.2], *p* < 0.01) [68]. Since the flow rate was similar in both groups, the authors concluded that a high P_v-a_CO_2_ might reveal the presence of a microcirculatory dysfunction. Regardless of the subtle difference in P_v-a_CO_2_ between groups, the study showed a correlation between P_v-a_CO_2_ and flow rate. Moreover, venous oxygen saturation and lactate were higher and hemoglobin was lower in nonsurvivors than in survivors. In the absence of direct microvascular assessment, differences in P_v-a_CO_2_ could be completely explained by these findings. Consequently, any reference to microcirculatory dysfunction may be reasonable but also speculative.

Contrary to the intriguing findings and interpretations of those studies [66,68], a large body of evidence shows that P_v-a_CO_2_ and CO are correlated in septic shock [37,39,40,41,42,43,45,46]. Moreover, several studies showed that systemic and regional P_v-a_CO_2_ fail to reflect microvascular perfusion because they are dependent on systemic or regional flow, and not on microvascular perfusion. In an experimental model of septic shock, the administration of endotoxin initially induced a hypodynamic state with reductions in CO, superior mesenteric artery blood flow, and mucosal microcirculatory perfusion. This condition was indicated by the widening of systemic, regional, and tissue PCO_2_ gradients [60]. Fluid resuscitation increased CO and superior mesenteric artery blood flow but failed to improve villi microcirculation. Accordingly, systemic and intestinal P_v-a_CO_2_ normalized. In contrast, mucosal P_t-a_CO_2_ remained elevated as an expression of the persistent villi hypoperfusion [60] (Figure 5). In patients with septic shock, sublingual microcirculation was altered and red blood cell velocity was low regardless of the systemic hemodynamic pattern [69]. P_mv-a_CO_2_, however, was lower in patients with hyperdynamic shock (cardiac index ≥ 4.0 L/min/m^2^) than in patients with normal CO (7 ± 2 vs. 5 ± 3 mm Hg, *p* < 0.05) (Figure 6). Another study, performed in patients with septic shock, found that skin flow was correlated with the cutaneous P_t-a_CO_2_ and was a strong predictor of outcome. As an expression of the lack of coherence between systemic hemodynamics and microcirculation, skin perfusion did not correlate with CO, and neither CO nor P_mv-a_CO_2_ was a predictor of outcome [62]. Unrelated to P_v-a_CO_2_, P_t-a_CO_2_ does track changes in microvascular perfusion [60,61,62].

## 3. Venous Minus Arterial Carbon Dioxide Partial Pressure to Arterial Minus Venous Oxygen Content Difference Ratio

### 3.1. Physiological Background

Under aerobic conditions, progressive workloads of exercise are associated with equivalent rises in VCO_2_ and VO_2_ as a reflection of the increasing oxidative metabolism. Therefore, the slope of the relationship—the RQ—persists initially unchanged. When the exercise becomes anaerobic, however, the increases in VCO_2_ surpass those from VO_2,_ and the RQ abruptly increases. This phenomenon concurs with the occurrence of hyperlactatemia and is known as the anaerobic threshold [70]. In the other extreme of physiology, during oxygen supply dependence, the RQ sharply rises because the decreases in VO_2_ are higher than the falls in VCO_2_ [11,12,13,14]. VO_2_ and VCO_2_ fall as an expression of the reduction in oxidative metabolism. The lower decrease in VCO_2_ is explained by the appearance of anaerobic VCO_2_. In both situations, the anaerobic exercise and the critical reductions in O_2_ delivery, the anaerobic VCO_2_ results from the buffering by bicarbonate of anaerobically generated protons. Consequently, the increase in RQ highlights the ongoing global anaerobic metabolism. Regional RQ, calculated as C_v-a_CO_2_/C_a-v_O_2_, has also been used to determine the presence of tissue hypoxia [28,71]. In a landmark study in pigs with endotoxemic shock, the use of epinephrine—compared to norepinephrine—was associated with lower blood flow and a higher P_v-a_CO_2_, lactate-to-pyruvate ratio, and gastric C_v-a_CO_2_/C_a-v_O2 [71].

Of note, the evaluation of RQ and CO_2_ contents is further complicated by the dynamics of CO_2_ stores and the time required to reach an equilibrium after hemodynamic, ventilatory, or metabolic changes [72]. Despite the lack of complete steady-state conditions, changes in expired gases quickly provide an alert about hemodynamic and metabolic changes [11,12,13,14,70].

Even though the determination of RQ is an attractive method for the identification of global tissue hypoxia, the metabolic carts needed for its measurement are not usually available in intensive care units. Additionally, measurements of RQ are not reliable if a high inspired oxygen fraction is used [73]. For these reasons, a simplification of Fick’s equation adapted to CO_2_, the P_v-a_CO_2_/C_a-v_O_2_, was proposed as a substitute for RQ [70]. Thus, high values of P_v-a_CO_2_/C_a-v_O_2_ with a cutoff of 1.4 have been associated with hyperlactatemia and high mortality [74]. Furthermore, P_cv-a_CO_2_/C_a-cv_O_2_ has been repeatedly included as part of algorithms for the assessment of tissue oxygenation [15,65,75,76]. Nevertheless, the evidence for these recommendations is quite limited and of low quality.

The utilization of P_cv-a_CO_2_/C_a-cv_O_2_ as a surrogate for RQ and tissue oxygenation depends on the following statements. First, RQ is the ratio between VCO_2_ and VO_2_:
RQ = VCO_2_/VO_2_(1)

Considering Fick’s equation, the previous equation can be reformulated as:
RQ = CO × C_mv-a_CO_2_/CO × C_a-mv_O_2_(2)

Next, a similarity between mixed and central samples is taken:RQ = Q × C_cv-a_CO_2_/Q × C_a-cv_O_2_(3)

Then, the common factor (CO) is simplified in numerator and denominator:
RQ = C_cv-a_CO_2_/C_a-cv_O_2_(4)

Finally, C_cv-a_CO_2_ is replaced by P_cv-a_CO_2_, assuming that CCO_2_ and PCO_2_ are linearly correlated over the physiological range of CO_2_ content:
RQ = P_cv-a_CO_2_/C_a-cv_O_2_(5)

Unfortunately, some of these expectations are problematic. In the following paragraphs, these questions will be discussed.

### 3.2. Limitations of P_cv-a_CO_2_/C_a-cv_O_2_ as a Surrogate of RQ

(1) The use of P_cv-a_CO_2_ instead of C_cv-a_CO_2_ in the calculation of the ratio: The investigators that proposed the utilization of P_cv-a_CO_2_/C_a-cv_O_2_ as a surrogate of RQ stated that given the almost linear relationship between CO_2_ content and partial pressure over the physiological range, P_cv-a_CO_2_ is an estimate of C_cv-a_CO_2_ in clinical practice [76]. As extensively discussed in the previous section, this asseveration is unsupported. Alterations in the CO_2_Hb dissociation curve, such as those induced by acidosis, hemodilution, and the Haldane effect, can substantially change the P_cv-a_CO_2_/C_a-cv_O_2_, regardless of the absence of alterations in RQ and tissue oxygenation. In septic patients, hyperoxia increases P_cv-a_CO_2_/C_a-cv_O_2_ from 2.63 ± 1.00 to 4.34 ± 3.37 (*p* < 0.03) despite the lack of changes in systemic hemodynamics and sublingual microcirculation [54]. An experimental study focused on the drawbacks of P_cv-a_CO_2_/C_a-cv_O_2_ as a surrogate for RQ [29]. P_mv-a_CO_2_/C_a-mv_O_2_, RQ, and their determinants were assessed during decreases in DO_2_ produced by stepwise bleeding or hemodilution. P_mv-a_CO_2_/C_a-mv_O_2_ and RQ were poorly correlated. Furthermore, in hemodilution, P_mv-a_CO_2_/C_a-mv_O_2_ increased even before the beginning of the oxygen supply dependence and the rise in RQ (Figure 5). This result was explained by the opposing effects of the decrease in Hb concentration on P_mv-a_CO_2_ and C_a-mv_O_2_. The former increased because of the reduced ability to carry CO_2_ in anemia while the latter decreased as occurs when the reduction in DO_2_ depends on the fall in arterial oxygen content (Figure 7). Additionally, in the last stage of DO_2_ reduction and despite comparable levels of anaerobic metabolism and increases in RQ, P_mv-a_CO_2_/C_a-mv_O_2_ markedly increased in hemodilution, compared to hemorrhage, because of the abovementioned reasons. Finally, Hb, metabolic acidosis, the Haldane effect, the position in a flattened portion of the CO_2_ dissociation curve, and RQ were found to be independent predictors of P_mv-a_CO_2_/C_a-mv_O_2_ in a multiple linear regression model. Although P_cv-a_CO_2_/C_a-cv_O_2_ was dependent on RQ, this was its weakest determinant [29]. Similar results were obtained during hypoxic hypoxia in a model of isolated hindlimb [77]. In this study, during progressive tissue hypoxia induced by hypoxemia or ischemia, P_va_CO_2_/C_av_O_2_ was disproportionally higher in hypoxic than in ischemic hypoxia (almost three times in the last stage) despite similar degrees of oxygen supply dependence. Moreover, P_va_CO_2_/C_av_O_2_ was higher in hypoxic than in ischemic hypoxia even before the beginning of the anaerobic metabolism.

P_cv-a_CO_2_/C_a-cv_O_2_ has been suggested as a tool to identify the aerobic or anaerobic origin of lactate [75,78]. As previously discussed, lactic acidosis can increase P_cv-a_CO_2_/C_a-cv_O_2_ because of its effects on the binding of CO_2_ to Hb, regardless of the aerobic or anaerobic production of lactate. In an experimental model of hemorrhagic shock, blood retransfusion normalized VO_2_ and RQ, but P_mv-a_CO_2_/C_a-mv_O_2_ remained high as a probable consequence of persistent hyperlactatemia [79]. In view of that, P_v-a_CO_2_/C_a-v_O_2_ could be considered a misleading tool to establish the meaning of hyperlactatemia. Similar demonstrations are required in other settings such as septic shock before generalizing this concept.

(2) The poor agreement between central and mixed venous samples: Central and mixed venous blood samples are not interchangeable for the different calculations. Although a small study advocated that mixed venous and central O_2_ saturation have similar behavior [80], a multicenter study demonstrated that both variables have poor agreement and that the direction of their changes over time can be different [81]. The problem is even worse for CO_2_-derived variables. In a clinical study, the 95% limits of agreement between P_cv-a_CO_2_/C_a-cv_O_2_ and P_mv-a_CO_2_/C_a-mv_O_2_ were 1.48, which is clinically unacceptable [44].

(3) The use of a defined cutoff of P_cv-a_CO_2_/C_a-cv_O_2_ for the identification of the anaerobic threshold: Depending on the metabolic substrate used for oxidative metabolism, the normal RQ ranges from 0.67 to 1.30 [82]. Carbohydrate-based diet and overfeeding increase RQ while fat diet and fasting decrease RQ. In this way, the start of anaerobic metabolism is indicated by abrupt increases in RQ, not by a particular value [11,12,13,14]. The same consideration is valid for the P_cv-a_CO_2_/C_a-cv_O_2_.

(4) The use of calculated O_2_ saturation for P_cv-a_CO_2_/C_a-cv_O_2_: In some studies, the computation of P_cv-a_CO_2_/C_a-cv_O_2_ was performed by the use of O_2_ saturation calculated from blood gases and oxyhemoglobin dissociation curve instead of measurements by co-oximetry [66,83,84]. This is a severe methodological mistake because calculated O_2_ saturation is not a reliable estimate of measured values. In addition, the error of measurement is additionally propagated in the calculation of P_cv-a_CO_2_/C_a-cv_O_2_. Moreover, paired measurements of P_cv-a_CO_2_/C_a-cv_O_2_ in the same analyzer are poorly reproducible with 95% limits of agreement of 1.22 [59].

### 3.3. The Physiological Feasibility of Increased P_cv-a_CO_2_/C_a-cv_O_2_ as a Reflection of Tissue Hypoxia in Critically Ill Patients

In experiments on oxygen supply dependence, the raise in RQ is a sudden phenomenon leading to rapid death. In stepwise hemodilution, RQ rises only when Hb decreases to 1.2 g%. Similarly, in progressive hemorrhage, RQ increases when mean arterial pressure is lower than 30 mm Hg [10]. These are extreme and obvious conditions that can be easily diagnosed. High values of P_cv-a_CO_2_/C_a-cv_O_2_ in adequately resuscitated patients rarely express global anaerobic metabolism. In contrast, they almost certainly result from the occurrence of factors that alter the of CO_2_Hb dissociation curve, as shown in experimental models [29] and in high-risk noncardiac surgery [85]. In both circumstances, RQ and P_v-a_CO_2_/C_a-v_O_2_ showed a different behavior. In critically ill patients, a direct comparison between P_cv-a_CO_2_/C_a-cv_O_2_ and RQ has not yet been performed. Therefore, values of P_cv-a_CO_2_/C_a-cv_O_2_ should be cautiously interpreted in stable patients.

### 3.4. The Clinical Usefulness of P_cv-a_CO_2_/C_a-cv_O_2_

Despite the fact that P_cv-a_CO_2_/C_a-cv_O_2_ might not track the true value of RQ, it might still be useful to reflect the severity and predict the outcome of critical illness. Since it is partially determined by Hb and base excess, anemia, and metabolic acidosis can result in high P_cv-a_CO_2_/C_a-cv_O_2_ by themselves and highlight the presence of a severe condition or be predictors of mortality [86,87]. Thus, anemia and metabolic acidosis might be responsible for the predictive ability of P_cv-a_CO_2_/C_a-cv_O_2_.

The ability of P_cv-a_CO_2_/C_a-cv_O_2_ as a predictor of outcomes in critically ill patients has been extensively reviewed elsewhere [88]. More than twenty years ago, a retrospective study performed in 89 patients monitored with a Swan–Ganz catheter found that a value of P_mv-a_CO_2_/C_a-mv_O_2_ higher than 1.4 was a predictor of hyperlactatemia and mortality [74]. Yet, P_mv-a_CO_2_/C_a-mv_O_2_ values were similar in nonsurvivors and survivors (1.7 ± 1.0 vs. 1.3 ± 0.5). In contrast, lactate showed a better prognostic ability than P_mv-a_CO_2_/C_a-mv_O_2_ and was higher in nonsurvivors (5.4 ± 6.1 vs. 2.0 ± 1.5 mmol/L). Despite the fact that P_mv-a_CO_2_/C_a-mv_O_2_ and lactate were different over time in survivors and nonsurvivors, only C_mv-a_CO_2_/C_a-mv_O_2_ and lactate, but not P_mv-a_CO_2_/C_a-mv_O_2_, were predictors of outcome in 135 patients with septic shock [83]. In another study, P_cv-a_CO_2_/C_a-cv_O_2_ and lactate were lower in survivors than in nonsurvivors, but lactate was a better predictor of mortality (AUROC curves of 0.73 and 0.81, respectively) [89]. The combination of P_cv-a_CO_2_/C_a-cv_O_2_ and lactate was a better predictor of mortality and organ failures than each individual variable in a retrospective study that recruited 144 patients with septic shock [84]. Additionally, in 35 patients with septic shock, P_cv-a_CO_2_/C_a-cv_O_2_ was a strong predictor of lactate behavior, and both variables were associated with mortality [90]. Recent studies also found a relationship of P_cv-a_CO_2_/C_a-cv_O_2_ to mortality [91,92,93].

In contrast, other studies failed to find an association between P_cv-a_CO_2_/C_a-cv_O_2_ and lactate or outcome. In a large multicenter cohort study that included 363 patients with septic shock, P_cv-a_CO_2_/C_a-cv_O_2_ could not differentiate patients with hyperlactatemia or poor lactate clearance from patients with normal lactate levels or adequate lactate clearance [94]. Another observational study in 23 septic patients showed that P_cv-a_CO_2_/C_a-cv_O_2_ and P_mv-a_CO_2_/C_a-mv_O_2_ were similar in survivors and nonsurvivors [44]. In high-risk surgical patients, RQ was a predictor of postoperative complications whereas P_cv-a_CO_2_/C_a-cv_O_2_ showed no prognostic ability [85].

A recent systematic review and meta-analysis found that P_cv-a_CO_2_/C_a-cv_O_2_ is associated with outcome [85]. Although the study showed little or no difference in the ability of P_cv-a_CO_2_/C_a-cv_O_2_ and lactate to predict mortality, there was a trend favoring lactate. Nevertheless, the conclusions were limited by the considerable heterogeneity among the studies. After the publication of this meta-analysis, a large prospective observational study including 456 patients with septic shock compared the prognostic ability of lactate, P_cv-a_CO_2_, and P_cv-a_CO_2_/C_a-cv_O_2_ [95]. Lactate at 6 h had the best predictive ability (AUROC of 0.902, 0.791, and 0.793, respectively). The combination of lactate and P_cv-a_CO_2_ only resulted in trivial increases in the predictive value (AUROC = 0.930). In another recently published study in 98 patients with septic shock, P_cv-a_CO_2_/C_a-cv_O_2_ at 24 h, but not at 8 h, was higher in nonsurvivors than in survivors and was a predictor of lactate clearance [96]. In contrast, lactate clearance was associated with outcomes at 8 h and 24 h.

Even though the relationship between P_cv-a_CO_2_/C_a-cv_O_2_ and outcome is conflictive, high values of P_cv-a_CO_2_/C_a-cv_O_2_ have some prognostic implications. The ability to predict mortality, however, is not superior to that of lactate. There are also controversial results about the relationship between P_cv-a_CO_2_/C_a-cv_O_2_ and lactate.

P_cv-a_CO_2_/C_a-cv_O_2_ has also been used as a predictor of the dependence of VO_2_ on DO_2_ [43,97,98]. The oxygen supply dependence might indicate the occurrence of alterations in oxygen extraction and an oxygen debt, but its actual meaning is debatable [99]. Considering that VO_2_ and DO_2_ are usually computed from a common variable (CO), and the magnitude of change of the calculated variables is usually small, there is a considerable risk of mathematical coupling of data. Thus, oxygen supply dependence might not be an actual fact but an artifact. Moreover, those studies have a gross methodological drawback because VO_2_ was calculated using central venous instead of mixed venous samples. In other studies, however, P_cv-a_CO_2_/C_a-cv_O_2_ did not predict the increase in VO_2_ in response to a fluid challenge [100,101]. Therefore, the evidence regarding this issue is inconclusive.

The usefulness of P_cv-a_CO_2_/C_a-cv_O_2_ as a goal of resuscitation has only been assessed in two studies [47,102]. In a controlled trial, 228 septic patients were randomized to either P_cv-a_CO_2_/C_a-cv_O_2_ or central venous oxygen saturation-targeted resuscitation. Mortality, organ failures, length of stay, and other secondary outcomes were similar in both groups [102]. In another small, controlled study, P_cv-a_CO_2_/C_a-cv_O_2_ was not better than lactate as a goal for the resuscitation of septic patients [47].

## 4. Future Directions

The lack of correlation between P_v-a_CO_2_ and microvascular perfusion in states of normal/high CO needs to be additionally confirmed. New studies should comprehensively assess the microcirculation and the multiple determinants of P_v-a_CO_2_, including changes in hemoglobin levels, acid-base status, the Haldane effect, temperature, and ventilation. Clinical research using metabolic cards, in critically ill patients, should also confirm that P_cv-a_CO_2_/C_a-cv_O_2_ is poorly correlated with RQ. Furthermore, the clinical usefulness of RQ in the monitoring of critically ill patients has never been tested.

## 5. Conclusions

P_v-a_CO_2_ and P_t-a_CO_2_ are mainly determined by blood flow. According to Fick’s principle, P_mv-a_CO_2_ and P_cv-a_CO_2_ are correlated with CO in physiological conditions and in critically ill patients, even in those with septic shock. Nevertheless, the relationship between CO and P_v-a_CO_2_ is not straightforward because of the changes in the CO_2_ dissociation curve and in the metabolic VCO_2_. While there is a widespread belief that P_cv-a_CO_2_ reflects microvascular tissue perfusion, this point of view is only based on the controversial results of one observational study. The concept is mistaken because it overlooks basic physiological foundations, as well as a large body of experimental and clinical evidence. If systemic flow seems adequate, increases in P_mv-a_CO_2_ or P_cv-a_CO_2_ firstly indicate the presence of factors that increase the dissociation of CO_2_ from Hb, such as anemia, metabolic acidosis, and the Haldane effect. In contrast, P_mv-a_CO_2_ and P_cv-a_CO_2_ are indicators of tissue perfusion in low-flow states. Unlike P_v-a_CO_2_, P_t-a_CO_2_ does reflect microcirculatory perfusion. Unfortunately, no technology is available nowadays for the measurement of tissue PCO_2_.

The clinical use of P_cv-a_CO_2_/C_a-cv_O_2_ as a substitute for RQ is conflictive. First, the increase in RQ secondary to critical reductions in DO_2_ is a life-threatening and striking condition. It is an easily noticeable event, which does not probably require further monitoring. Given that the start of anaerobic metabolism is indicated by the sharp rise in the RQ, and the normal range of RQ is wide, the use of a defined cutoff of 1.4 for P_cv-a_CO_2_/C_a-cv_O_2_ is irrelevant. Moreover, P_cv-a_CO_2_/C_a-cv_O_2_ is more dependent on factors that modify the CO_2_Hb dissociation curve than on the actual RQ. Experimental studies showed that RQ and P_cv-a_CO_2_/C_a-cv_O_2_ might exhibit distinct behaviors in different models. The ability of P_cv-a_CO_2_/C_a-cv_O_2_ to predict the mortality of critically ill patients is not superior, but probably lower than that of lactate. In addition, the association with mortality could be related to the impact of acidosis and anemia on the ratio. Regardless of its meaning, the relationship of P_cv-a_CO_2_/C_a-cv_O_2_ to oxygen supply dependency is controversial. A randomized controlled trial also showed that P_cv-a_CO_2_/C_a-cv_O_2_ is useless as a goal of resuscitation in sepsis. The use of P_cv-a_CO_2_/C_a-cv_O_2_ as an index of tissue oxygenation lacks a physiological basis and solid evidence.

In brief, P_cv-a_CO_2_ and P_cv-a_CO_2_/C_a-cv_O_2_ are complex variables with multiple determinants. Accordingly, their interpretation requires careful analysis. The direct assumption that high values of P_cv-a_CO_2_ and P_cv-a_CO_2_/C_a-cv_O_2_ are signs of microcirculatory hypoperfusion and anaerobic metabolism should be avoided. P_cv-a_CO_2_ is a marker of cardiac output. In states of low cardiac output, increased P_cv-a_CO_2_ reflects global tissue hypoperfusion. In conditions of normal or high cardiac output, high values should be explained by changes in the two other determinants, the CO_2_Hb dissociation curve and the VCO_2_, and not by an altered microcirculation. Since the calculation of P_cv-a_CO_2_/C_a-cv_O_2_ is derived from the determinants of the RQ, it has been considered a surrogate for RQ and tissue oxygenation. Nevertheless, it is more dependent on factors that modify the dissociation of CO_2_ from Hb than on the actual RQ measured by analysis of expired gases. Therefore, high values should be interpreted with extreme caution.

## Figures and Tables

**Figure 1 medicina-59-01262-f001:**
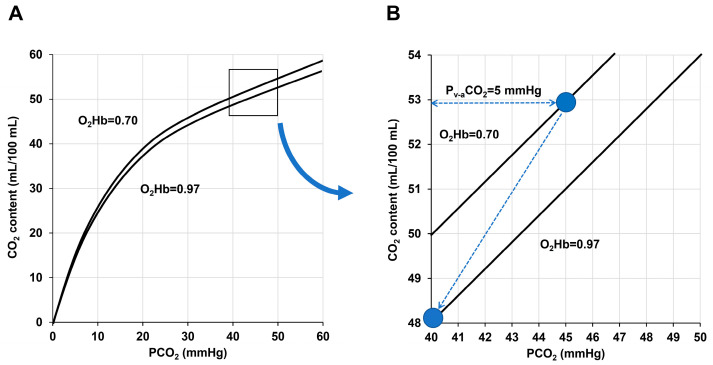
CO_2_Hb dissociation curve. (**A**): Oxygenated Hb has a lower affinity for CO_2_ and the curve has a right shift (Haldane effect). Metabolic acidosis and anemia produce displacement in the same direction. (**B**). Deoxygenated venous blood has a better ability to carry CO_2_ in the peripheral capillaries whereas the oxygenation of Hb in the pulmonary circulation enhances the alveolar elimination of CO_2_.

**Figure 2 medicina-59-01262-f002:**
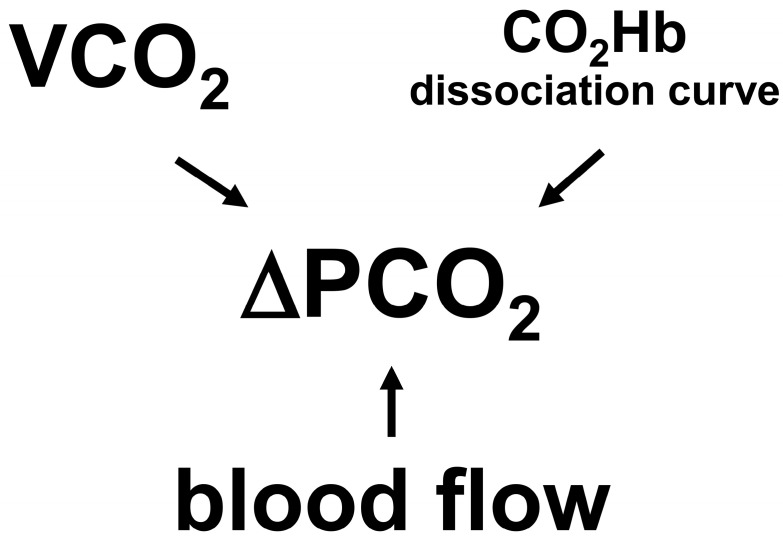
Determinants of venous minus arterial and tissue minus arterial PCO_2_ differences). Venous minus arterial and tissue minus arterial PCO_2_ differences (ΔPCO_2_) are the result of interactions among CO_2_ production (VCO_2_), CO_2_Hb dissociation curve, and blood flow. Isolated changes in any determinant can independently modify PCO_2_ differences.

**Figure 3 medicina-59-01262-f003:**
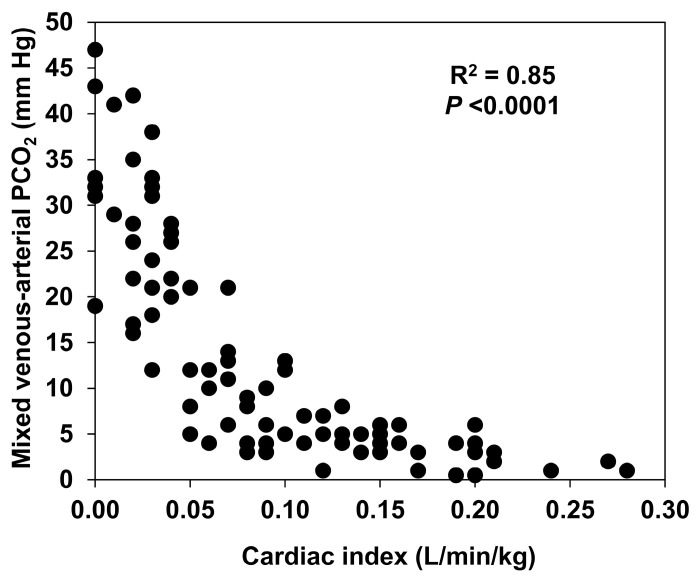
Correlation between cardiac output and mixed venous minus arterial PCO_2_ difference. The reductions in cardiac output induced by progressive bleeding are strongly associated with semilogarithmic increases in mixed venous minus arterial PCO_2_ difference. Reproduced from Ref. [34] with permission.

**Figure 4 medicina-59-01262-f004:**
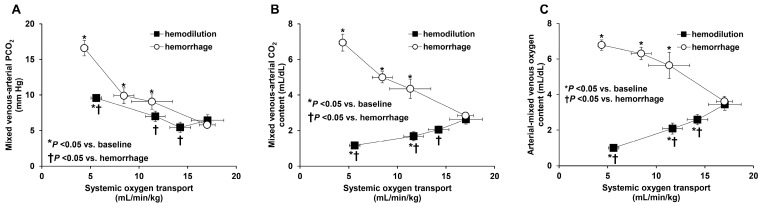
Relationship of systemic oxygen transport to mixed venous minus arterial PCO_2_ difference (**A**), mixed venous minus arterial CO_2_ content difference (**B**), and arterial minus mixed venous oxygen content difference (**C**) in sheep that underwent progressive bleeding or hemodilution. In hemorrhage, the three variables increased. In hemodilution, there were opposite changes in mixed venous minus arterial CO_2_ partial pressure and content difference (the former increased, and the latter decreased), whereas arterial minus mixed venous oxygen content difference decreased. Reproduced from Ref. [29] with permission.

**Figure 5 medicina-59-01262-f005:**
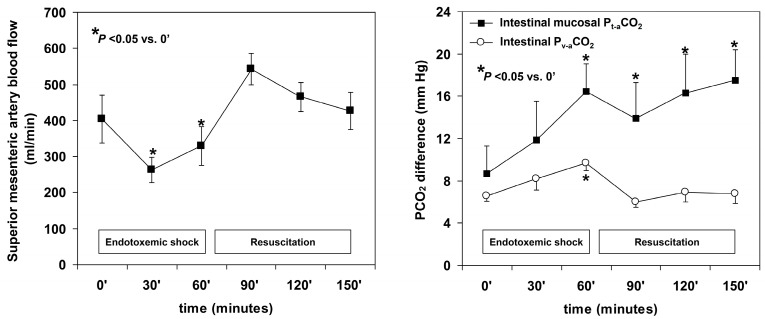
Failure of venous minus arterial PCO_2_ difference (P_mv-a_CO_2_) to reflect tissue perfusion in an experimental model of endotoxemic shock and fluid resuscitation. In experimental septic shock, the administration of endotoxin initially induced a hypodynamic state with reductions in cardiac output, superior mesenteric artery blood flow, and mucosal microcirculatory perfusion. This condition was indicated by the widening of systemic, regional, and tissue PCO_2_ gradients. Fluid resuscitation increased cardiac output and superior mesenteric artery blood flow but failed to improve villi microcirculation. Accordingly, systemic and intestinal venous minus arterial PCO_2_ difference (P_v-a_CO_2_) normalized. In contrast, mucosal tissue minus arterial PCO_2_ (P_t-a_CO_2_) remained elevated as an expression of the persistent villi hypoperfusion (From data of Ref. [60]).

**Figure 6 medicina-59-01262-f006:**
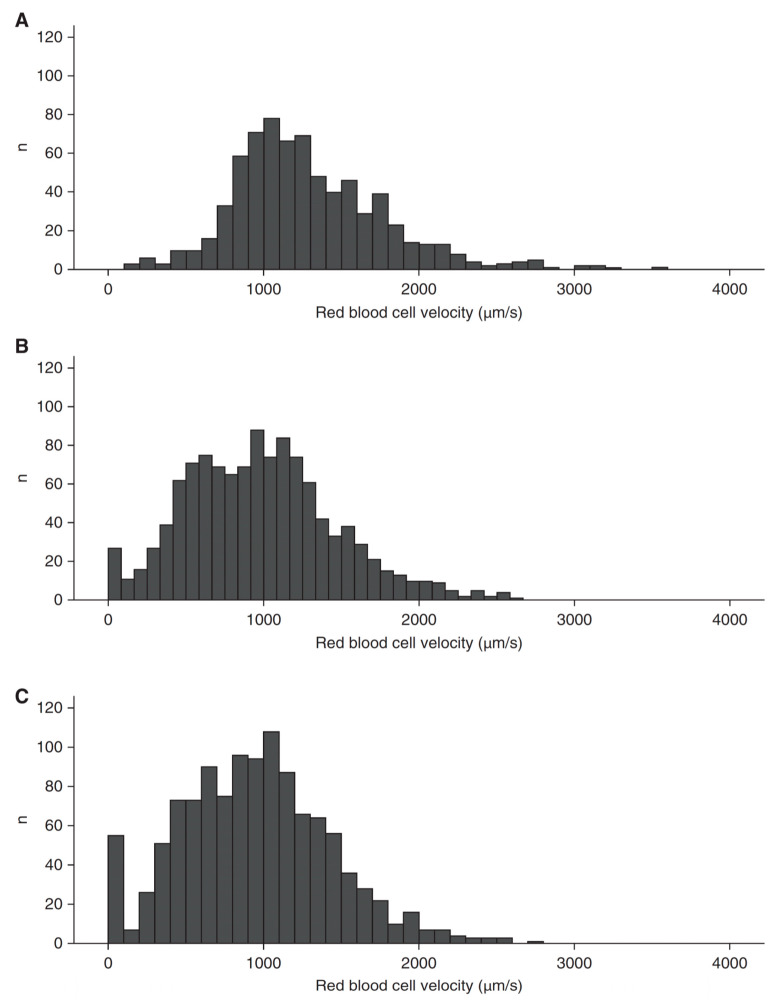
Histograms of sublingual red blood cell velocities. (**A**): Healthy volunteers. (**B**): Patients with normodynamic septic shock (cardiac index = 2.55 ± 0.43 mL/min/m^2^). (**C**): Patients with hyperdynamic septic shock (cardiac index = 4.90 ± 0.91 mL/min/m^2^). The histograms of patients with normo- and hyperdynamic septic shock were similar and shifted to the left (lower velocities). Nevertheless, the mixed venous minus arterial PCO_2_ difference was higher in normo- than in hyperdynamic patients (7 ± 2 vs. 5 ± 3 mm Hg, *p* < 0.05). Reprinted from Ref. [69] with permission of the American Thoracic Society. Copyright © 2023 American Thoracic Society. All rights reserved.

**Figure 7 medicina-59-01262-f007:**
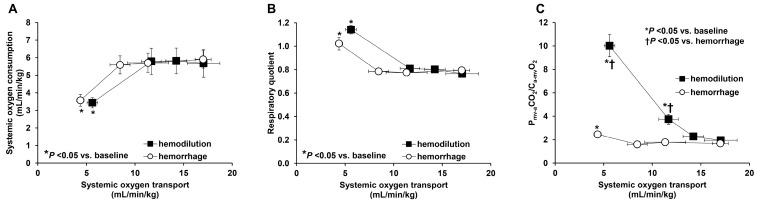
Relationship of systemic oxygen transport to systemic oxygen consumption (**A**), respiratory quotient (**B**), and the ratio of mixed venous minus arterial PCO_2_ difference to arterial minus mixed venous oxygen content difference (P_mv-a_CO_2_/C_a-mv_O_2_) (**C**) in sheep that underwent progressive bleeding or hemodilution. There were similar degrees of oxygen supply dependence and increases in the respiratory quotient in both groups. In hemodilution, however, the elevation in P_mv-a_CO_2_/C_a-mv_O_2_ was disproportionately higher than in hemorrhage and developed even before the development of anaerobic metabolism. Reproduced from Ref. [29] with permission.

## Data Availability

Not applicable.

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
