# Peer review of "Venous Minus Arterial Carbon Dioxide Gradients in the Monitoring of Tissue Perfusion and Oxygenation: A Narrative Review"

_medicina, 2023, doi:10.3390/medicina59071262_

Round 1

Reviewer 1 Report

This is a timely and very important narrative review - Prof. Dubin is an outstanding expert in the field. The paper is well written and clearly structured. I have no major requests for revisions of this well-reading text.

There is one important issue, though, that I would like to be addressed on the limitation of the CO2-contents to estimate the RQ. Effect of changes in the body CO2-pool should be clearly addressed in the limitations of any RQ-assessment. Any change in metabolic rate, substrate utilization, breathing pattern, and/or settings of mechanical ventilation (changes in alveloar ventilation) will induce changes in the body pool of CO2. The achievement of a new steady state is slow and during this time the blood CO2-content will not reflect the true RQ – in practice, metabolic cart measurements suggest that 60-90 minutes will be necessary to get a reasonable estimate of the VCO2. Please add a comment of this in the limitations.

Some relevant papers on the CO2-contents vs PCO2 should be added – please consider the following:

Martikainen TJ, Tenhunen JJ, Giovannini I, Uusaro A, Ruokonen E. Epinephrine induces tissue perfusion deficit in porcine endotoxin shock: evaluation by regional CO(2) content gradients and lactate-to-pyruvate ratios. Am J Physiol Gastrointest Liver Physiol. 2005 Mar;288(3):G586-92. doi: 10.1152/ajpgi.00378.2004. Epub 2004 Oct 28. PMID: 15513952.

Cavaliere F, Giovannini I, Chiarla C, Conti G, Pennisi MA, Montini L, Gaspari R, Proietti R. Comparison of two methods to assess blood CO2 equilibration curve in mechanically ventilated patients. Respir Physiol Neurobiol. 2005 Mar;146(1):77-83. doi: 10.1016/j.resp.2004.11.008. PMID: 15733781.

Chiarla C, Giovannini I. Blood CO2 exchange monitoring, Haldane effect and other calculations in sepsis and critical illness. J Clin Monit Comput. 2019 Apr;33(2):357-358. doi: 10.1007/s10877-018-0160-1. Epub 2018 May 25. PMID: 29802513.

No comments.

Reviewer 2 Report

I have made comments and suggestions in my review

Reviewer 3 Report

I read with great interest the manuscript by Dubin and Pozo on venous minus arterial carbon dioxide gradients in tissue perfusion and oxygenation monitoring. The authors performed an interesting and complete review from the physiological background to the future directions. The paper is well written and the topic is relevant. I have only minor issues to address.

Introduction

- When authors refer to capillary refill time, besides the limits of its assesment, they should also underline that it is an easy technique, that can be performed in different districts such as the finger tip, the earlobe, the forehead, the chest wall etc. (doi: 10.1186/s12871-022-01920-1 - doi: 10.1001/archpedi.1991.02160030064022). Please discuss and add these 2 references.

- Assessment of sublingual microcirculation is also a valid method to monitor tissue perfusion (doi: 10.1097/MCC.0000000000000495), although so far it is mainly used for research purposes. Please discuss and provide adequate reference.

Round 2

Reviewer 2 Report

The authors have adequately addressed or defended each of the concerns and suggestions raised in my review. Despite the fact that I may still disagree somewhat to their overall conclusion I believe they have now adequately presented their argument, and the purpose of scientific discourse is to at least be able to defend your point of view and for that view then to be available for all to interpret. I thank the authors for their detailed response